# Fantastic Foods and Where to Find Them—Advantages and Disadvantages of Nutri-Score in the Search for Healthier Food

**DOI:** 10.3390/nu14224843

**Published:** 2022-11-16

**Authors:** Dariusz Włodarek, Hubert Dobrowolski

**Affiliations:** 1Department of Dietetics, Institute of Human Nutrition Sciences, Warsaw University of Life Sciences (SGGW), 159c Nowoursynowska Str., 02-776 Warsaw, Poland; 2Department of Functional and Organic Food, Institute of Human Nutrition Sciences, Warsaw University of Life Sciences (SGGW), 159c Nowoursynowska Str., 02-776 Warsaw, Poland

**Keywords:** Nutri-Score, front-of-pack label, nutritional habits, nutritional choices, healthy eating

## Abstract

The composition and quality of food rations and the intake of nutrients with the diet are undoubtedly reflected in our health. In order to help the consumer choose the right food product, food manufacturers use front-of-pack labels, which are designed to convey concise information about the nutritional value of the product. Such labels include the Nutri-Score system. In this paper, we have critically analyzed this system. As the available data indicate, this system does not take into account a number of factors that affect the quality of a food product, including the size of the package, the contents of vitamins, minerals, and other selected health-promoting ingredients, the degree of processing, or the fatty acid profile of the product, and it discriminates against regional products, organic products, and juices and nectars. This system, although intuitive and created with good intentions, still has quite a few flaws that must be addressed before it can be considered to correctly indicate the nutritional value of food products.

## 1. Introduction

Proper diet quality is reflected in health and nutritional status. The proper nutritional value of foods consumed, supporting the ability to meet energy and nutritional needs, is a key element in terms of the composition of a well-balanced ration, which is of great importance in terms of the overall quality of the diet. Considering the impact of diet on the risk of diet-related diseases [1], there has been a clear trend in the last decade related to adopting healthier lifestyles, focusing on healthy eating, and leading an active lifestyle [2]. With the growing interest in how food affects health, consumers are looking more closely at food labels for nutritional information. Appropriate appearance and information on the packaging, therefore, play a huge role in improving the quality of the public’s diet, as well as the prevention of disease.

Front-of-pack (FoP) nutritional indicators play an important role in helping consumers make healthier food choices [3]. As highlighted by the World Health Organization (WHO), front-of-pack supplemental food labeling systems provide an additional valuable source of information regarding the nutritional value of food [4]. The concept of nutritional indicators is based on the premise that a properly constructed indicator is intended to provide the consumer with reliable knowledge regarding the nutritional value of food, as well as concise information regarding the potential health effects of consuming a given product. Indicators placed on packaging are intended to provide information about the nutritional value of a particular product. Placing such an indicator on the front of the package helps to simply and quickly identify the quality of a particular product and its translation into consumer health. One such indicator is the Nutri-Score system. 

Any system that is intentionally designed to improve food selection based on a specific algorithm is not free of flaws. It is limited by the range of parameters included in the algorithm, the number of which does not take into account the variety of food components that affect its overall and actual nutritional value. Moreover, it can be used by food manufacturers as a marketing tool, while lack of use may be a source of interpretation errors by consumers. This is no different in the Nutri-Score system. Hence, the purpose of this paper is to critically analyze the Nutri-Score system as a front-of-pack (FoP) labeling system.

## 2. The Nutri-Score System

Developed in France, Nutri-Score is a labeling system for the quality and health value of food products. It takes into account the contents of energy, simple sugars, saturated fat, and sodium (as negative components of the product), the percentage of fruit and vegetables, and the amount of fiber and protein (as positive components) [5]. In addition, according to the 2021 update [6], the addition of nuts, legumes, and canola, walnut, and olive oils are taken into account in the product assessment as positive elements. Points are awarded on the basis of the content of negative components of the product, while points are deducted for the content of positive components. On this basis, a food product can achieve a score from −15 to 40, which is used to provide a color coding (dark green to dark orange) and a letter coding (A to E), indicating the nutritional value of the product. It is worth noting that the Nutri-Score system evaluates the positive and negative components per 100 g of product. In addition, it should be pointed out that the way in which drinks, cheese, fats, and other products are assessed is done on the basis of separate, individual rules.

Various studies analyzing food composition tables from eight European countries (the EUROFIR database [7] and a large product database covering the food markets of 13 European countries [8]) indicate that for all the European countries evaluated, the Nutri-Score classification of foods was in line with public health nutrition recommendations: most products containing mainly fruits and vegetables are classified into category A or B, while most sweet products and salted snacks, sauces, and animal fats belong in D or E. The method of classification has also been confirmed in individual food groups: in the starchy products group, legumes, pasta, and rice are rated more favorably than breakfast cereals; in the dairy group, milk and yogurt are rated better than cheese. As for beverages, most fruit juices are rated C, carbonated beverages are rated E, and water is rated A [9]. 

The theoretical assumptions of the Nutri-Score system are very much in line with the dietary recommendations for a healthy diet issued by the WHO [10] and partly with those issued by the U.S. Department of Agriculture (USDA) (2020) [11] and the European Society of Cardiology [12]. However, it is not a substitute for general public health recommendations, especially dietary guidelines based on the consumption of food groups aimed at achieving a healthy diet for consumers. In addition, and necessarily emphasized, the dietary recommendations emphasize the consumption of unprocessed or minimally processed foods and limiting the consumption of ultra-processed foods, as well as the regular consumption of meals with an appropriate composition.

## 3. The Nutri-Score System in Light of Research

Food labeling with the Nutri-Score system has also been the subject of many studies. A study conducted by Donat-Vargas et al. in 2021 on a group of more than 12,000 participants found that the consumption of low-nutritional value products with low Nutri-Score gradations (D or E rating) was associated with higher mortality among participants (over a nearly 9-year follow-up period) in Spain. In addition, the results indicated a reduced risk of death when these products were substituted with fresh foods [13]. It is worth noting, however, that the evaluation concerned the diet of the subjects obtained on the basis of dietary history, and then the products were evaluated according to the Nutri-Score system. Thus, it was not the labeling according to this system that influenced consumer choices, but this was assessed retrospectively. Moreover, in 2008–2010, when dietary data were collected, the Nutri-Score system was not yet in use (the Nutri-Score system was introduced in 2017).

Furthermore, a study by Dréano-Trécant et al. (2020) demonstrated the effectiveness of the Nutri-Score system in supporting healthier dietary choices among study participants. As the authors point out, the Nutri-Score would be an appropriate tool for differentiating the nutritional value of products within and between relevant food groups in different European countries, while maintaining consistency with dietary recommendations [14]. 

Julia et al. (2015) [15] found that, overall, the distribution of food items for the French market in categories similar to the Nutri-Score (5-color nutrition label; 5-CNL), on which this indicator is based and whose scale was also based on five colors and a gradation system from A to E, was in line with French recommendations: 95.4% of “Fruits and vegetables” and 72.5% of “Cereals and potatoes” were classified as “Green” or “Yellow”, while 86.0% of “Sweet snacks” were classified as “Pink” or “Red”. For beverages, added fats, and cheeses, adjustments to the Food Standards Agency’s original scoring calculation model were necessary to comply with the official French dietary recommendations. Similar results were obtained for the UK market [16].

Subsequent studies have shown that people with a dietary profile corresponding to a better index according to profiling compliant with the Food Standards Agency Nutrient Profiling System (FSA NPS) have higher intake of fruits, vegetables, and fish, lower intake of sugary, fatty, and salty snacks, higher intake of fiber, vitamin C, beta-carotene, calcium, zinc, and iron, lower intake of saturated fatty acids, better adherence to public health dietary recommendations, and more favorable antioxidant profiles (higher blood levels of vitamin C and beta-carotene) [17].

There are also promising findings on the association of the individual dietary index (corresponding to the nutrient profiling system) with health outcomes. Food consumption based on the Food Standards Agency nutrient profiling system (FSA NPS) was associated with a lower risk of developing chronic diseases, including cancer, cardiovascular disease, weight gain, and metabolic syndrome. In addition, Nutri-Score labeling positively impacted the overall nutritional quality of shopping carts, with Nutri-Score performance outperforming all other labels tested [9].

There have also been a number of studies comparing the Nutri-Score system with other current label systems. In their work, Finkelstein et al. (2019) [18] demonstrated the superior performance of the Nutri-Score in shaping an appropriate diet compared to the English Multiple Traffic Lights (MTL) system and a control group. The MTL system provides the main nutrient information with individual color-coded assessments for each nutrient based on the reference intake and guidance provided by the European Food Information Council [18]. However, it is noteworthy that the number of calories consumed and sugars from beverages were not significantly reduced with Nutri-Score, in contrast to MTL. This suggests a better performance of the MTL label in obese individuals and those with metabolic diseases when the energy value of the ration needs to be reduced and/or the content of simple sugars in the diet needs to be reduced, compared to Nutri-Score and packages without nutritional indicators.

A study by Egnell et al. (2018) [19] compared the ability to understand nutritional information for five indicators: the Health Star Rating system (HSR), Multiple Traffic Lights (MTL), Nutri-Score, Reference Intakes (RIs) and the Warning symbol. The study was conducted on a group of 1000 participants from 12 countries, who were asked to sort products (a set of three pizzas, a set of three cakes, and a set of three breakfast cereals) based on the front-of-package labeling (FoPL) used, from best to worst. Across all countries and food categories, the result for the Nutri-Score indicator was the best. However, it should be noted that three product groups were used for the evaluation, and the effectiveness score was determined within the groups, not among them. Therefore, it is not possible to say to what extent the choices will translate into an overall dietary pattern. In addition, heavily processed products were selected for the study, and the results could have been significantly different if low-processed products were included in the evaluation. Moreover, respondents’ good understanding of labeling is not necessarily related to a better understanding of the nutritional value of products, which is crucial when not all products are labeled uniformly, but only about the ability to rank colors.

## 4. System Flaws and Consumer Choices

The Nutri-Score system was intentionally created with good intentions and with a mission necessary in an era of epidemic obesity and chronic non-communicable diseases induced by poor eating habits. As numerous studies have indicated, it largely fulfills its purpose. However, while numerous advantages can be cited, equally numerous disadvantages cannot be overlooked, which can result in negative behavior among consumers and in them making the wrong dietary choices. Indeed, bypassing the limitations of the methodology used in the Nutri-Score study may result in a reduction in the consumption of products or groups of products that, despite their low gradation according to the Nutri-Score, can at the same time be a very valuable component of a well-balanced diet both nutritionally and sensorially, and in the nutrition of groups of people of different ages. The following analysis was carried out using food products present on the Polish market and currently on sale, marked with the Nutri-Score system.

### 4.1. Nutri-Score as an Algorithm per 100 g or 100 mL of Product

The algorithm for the Nutri-Score labeling system involves determining the contents of selected ingredients and energy per 100 g of product. The final approach of consumers to the consumption volume of products labeled with particular colors is not known. The labeling is on packages of various sizes—from small, such as 50 g, to large, such as 1000 g. No consideration is given to the typical portion size of products consumed. Thus, there is a risk that similar products from the higher Nutri-Score category will generally be consumed in larger amounts, and products in large packages with a label from the higher Nutri-Score category may be consumed without any thought that they should be consumed in a certain amount, and the label refers to the amount per 100 g. Ultimately, this can paradoxically increase the energy value of the diet and the supply of health-adverse nutrients.

A study by van den Akker et al. (2022) indicated that the Nutri-Score promotes healthier dietary choices, but this did not affect package size selection [20]. This means that choosing a healthier product may translate into consuming more of the product in a larger package size, and thus, consuming more of the ingredients that Nutri-Score distinguishes as negative. Table 1 shows two cream cheeses. A consumer, wanting to eat healthily, would likely choose a product with a better Nutri-Score gradation while disregarding the package size. Since the consumption volume of a product in such packages is usually equivalent to the size of the package, it is reasonable to assume that a consumer, choosing a more favorably labeled product, would consume more energy, saturated fatty acids, and simple sugars as a derivative of the amount consumed than if they chose a product with a lower gradation and consumed the appropriate amount. Also of note, in the case of Nutri-Score, the amount of energy consumed and sugars from beverages were not significantly reduced [18], which may be due to misinterpretation in the context of package size.

In addition, Nutri-Score’s failure to distinguish between the size of a consumed serving of a product (or the size of a package) and only converting to 100 g or 100 mL of a product may lead to a reduction in the consumption of products with inferior gradations, which are customarily consumed in smaller quantities (Table 2). Looking only at the Nutri-Score overlooks the fact that camembert cheese is consumed in much smaller quantities than a whole package, and feta cheese serves as an addition to other dishes (mainly salads) and is also consumed in much smaller quantities. Considering the size of a customary serving of these products, the amount of individual nutrients ultimately supplied is therefore considerably less than what is implied by the conversion per 100 g/mL of product. In addition, the amount of potentially negatively impacting nutrients delivered along with the suggested small portion size does not pose a significant health risk to the consumer. Consuming a small serving of a product with a lower gradation may even be healthier than consuming a larger serving of a product with a better gradation. As is well known, the size of the consumption of a particular product translates into the supply of nutrients both favorably and negatively affecting health.

### 4.2. Nutri-Score vs. Vitamins and Minerals

The Nutri-Score system does not take into account values such as the content of vitamins and minerals and other bioactive ingredients. Products that are valuable sources of these nutrients can often receive a lower Nutri-Score gradient than they actually deserve. As presented in Table 3, products with a higher nutritional value due to their significantly higher content of vitamins, minerals, and other bioactive compounds are assigned a lower Nutri-Score due to the high content of monosaccharides present in them (in the case of juices, resulting from their content in the raw material from which they were made, and not subject to technological modification due to legislation on their production). In addition, black currant nectars have higher contents of vitamins and minerals per serving despite the lack of significant differences in energy value compared to apple juice (a difference of 8 kJ per 100 mL). However, they receive a worse Nutri-Score. For comparison, a cola-type beverage is presented that, despite lacking any nutritional value, receives a better Nutri-Score simply because it contains no added sugar, despite containing other anti-nutritional compounds such as phosphoric acid and sweeteners, the long-term health effects of which are still not established, and new reports indicate that they may be adverse. According to the results from the EPIC study, the consumption of artificially sweetened soft drinks was associated with increased mortality from any cause [21]. The ESPGHAN Nutrition Committee [22,23] points out that the effects of sweeteners on long-term metabolic health are currently not well understood, and there is little evidence available to develop an evidence-based recommendation for their use in children and adolescents. In addition, the Polish Society of Pediatric Gastroenterology, Hepatology and Nutrition [24] notes that artificially sweetened beverages should not be used in children’s nutrition.

### 4.3. Nutri-Score vs. Fruit and Vegetable Juices

In addition, special attention should be paid to the Nutri-Score labeling of juices. As mentioned, juices contain sugar derived from the content in the original product, i.e., from fruits or vegetables, with lower protein and dietary fiber content, which means that the Nutri-Score will oscillate between C and D depending on their nutritional value. Given the content of many vitamins and minerals, as well as many other bioactive compounds, this rating is misleading, especially since artificially sweetened beverages without any nutritional value receive a B grade. It is worth pointing out that, according to the HAPIEE study conducted by Garduño-Alanís et al. (2019), the consumption of both sugar-sweetened and artificially sweetened non-alcoholic beverages was positively correlated with the BMI of a study group of 26,634 participants from Eastern Europe, while no binding association was obtained for juices [26].

The presence of juices in the diet can have a positive impact on diet quality. As demonstrated in a study by O’Neil et al. (2011), consumption of 100% orange juice was associated with better diet quality and better nutrient selection, with no increase in the risk of obesity and overweight in children [27]. This study, therefore, demonstrates that the presence of juices in the diet leads to an effect, which is the premise of the Nutri-Score index in itself. This makes the low score for juices and nectars all the more incomprehensible, while these products fit into the goals and objectives of this system. Similar results were obtained by Sakaki et al. (2019). The results achieved in this study involving 26,554 children and adolescents from the United States showed a reduction in the prevalence of obesity among boys consuming orange juice (with no significant differences for girls), and a clear tendency for children consuming orange juice to practice healthier eating habits and healthy lifestyles, with no increased prevalence of overweight or obesity [28]. Rampersaud and Valim (2017) report that moderate consumption of citrus juices can provide dietary improvement and does not appear to negatively affect body weight and composition or other anthropometric indices in children and adults [29]. In addition, a study by Ribeiro et al. (2017), conducted with 78 obese adult patients, indicated that a low-energy diet combined with orange juice consumption contributed to weight loss and improved diet quality and obesity-related metabolic outcomes to a greater extent than in control patients who consumed only a low-energy diet without the juice. Moreover, orange juice was shown to increase insulin sensitivity, improve the lipid profile, and exhibit anti-inflammatory effects [30]. Orange juice is not the only one to show such properties. A study by Babaeian et al. (2013) with type 2 diabetes patients showed that consumption of unsweetened pomegranate juice improved insulin sensitivity, reduced body weight and waist and hip circumference [31].

Adequate dietary intake of juices, embedded in the dietary pattern, is additionally beneficial not only for improving anthropometric parameters and improving insulin sensitivity. As indicated by the results of numerous studies, regular consumption of juices also translates into a preventive and protective effect against many diseases. For example, a study by Foschi et al. (2010) involving a representative group of oncology patients with various cancer sites showed that citrus fruits have a protective role against cancers of the gastrointestinal tract and upper respiratory tract [32]. Jaganathan et al. (2014) point out that the unique biochemical composition of fruit juices influences their anticancer properties [33,34]. On the other hand, a review study by Ruxton et al. (2006) clearly showed that there is no evidence showing that drinking fruit and vegetable juices is less effective in preventing cancer and coronary heart disease compared to consuming whole fruits and vegetables [35]. Many studies indicated that regular consumption of juices has a beneficial effect on the serum lipid profile [36,37,38], antioxidative status, and oxidative stress [36,37,38], reduced the risk of cardiovascular diseases [36,39], and indicated a preventive role in metabolic syndrome risk reduction [38]. 

Juices in the daily diet have also shown potential effects on neurodegenerative diseases. Dai et al. (2006) observed that consumption of fruit and vegetable juices, which contain high amounts of polyphenols, reduces the risk of Alzheimer’s disease [40]. Other researchers have observed that consumption of orange juice improves cognitive function in middle-aged and elderly people [41,42], and apple juice improves function in people with Alzheimer’s disease [43]. There are even reports available indicating that tomato juice administration may have a moderating effect on the course of bronchial asthma [44]. 

The health effects of fruit juice may be summarized by the insights from a recent review publication by Ruxton and Myers (2021) [45]. The authors concluded that consumption of 100% fruit juices provides, in addition to carbohydrates, also micronutrients (including potassium, vitamin C, folate, carotenoids, and other vitamins) and other bioactive compounds (such as polyphenols: hesperidin, quercetin, carotenoids, anthocyanins, and punicalagin) at levels similar to their occurrence in whole fruit. The results showed that consuming these juices at levels consistent with dietary guidelines, i.e., 75–224 mL per day (depending on the age group), does not increase the risk of developing obesity, type 2 diabetes, cardiovascular disease, or worsening glycemic control. On the contrary, regular consumption of juices appears to have health benefits in terms of vascular function, lower blood pressure, reduced risk of stroke, and improved cognitive function. Observational studies have reported associations between the presence of juices in the diet and an adequate supply of nutrients. In summary, the authors concluded that fruit juices appear to offer more health benefits than risks, and there appears to be no justification for discouraging their consumption by children, adults, or the elderly as part of a balanced diet [45].

### 4.4. Nutri-Score vs. Degree of Food Processing

The algorithm adopted in the Nutri-Score means that products with different utilities in a properly balanced diet, compliance with dietary recommendations, and potential for use in dietary therapy for various conditions, degrees of processing, or health effects can receive the same score. 

An example is grain products. Pasta products have such a favorable nutritional value that, regardless of the type of flour used in production, they will receive the best possible Nutri-Score index, and thus, the message to the consumer is that there is no difference between whole grain pasta and pasta prepared from ordinary white flour. A lack of proper education on how to select a product according to the degree of processing or selected ingredients (not included in the constituent score of the index), and relying only on the summary color index, may not be sufficient for the consumer in making a good choice of a food product recommended for proper nutrition, and thus, may not be reflected in improved diet quality.

It is worth noting that, based on a study of the diet quality of children, adolescents, and adults living in Europe (study by I. Family) [46], it was found that energy intake from ultra-processed foods accounted for almost half of the subjects’ daily energy intake. This was mainly due to the consumption of calorie-rich foods with low nutritional value. Therefore, there is a risk that simplified labeling using the Nutri-Score may influence not only the promoted consumption of foods with a lower energy value per 100 g of product, but also products with a high degree of processing and in which the composition has been modified to obtain a better rating gradient, while discouraging the choice of low-processed foods, whose composition cannot be modified and which have obtained a lower gradient.

Other good examples highlighting this issue are selected cereal products. Rice and groats will receive the highest index in the Nutri-Score system, while there are significant differences in the parameters not included in their evaluation, that affect their potential for use in the diet and in selected disease entities (such as diabetes). Examples include products such as white rice, parboiled rice, basmati rice, or jasmine rice, which obtained the same value in the Nutri-Score index, but differ in their values of the glycemic index, among others (Table 4). 

### 4.5. Nutri-Score vs. Type of Fatty Acids in Product Formulation

Failure to account for the type of fatty acids supplied with fat in the Nutri-Score system can result in the misconception that products with a higher fat content are less healthy than those with a lower fat content. Fat is not only an energy component and a source of saturated fatty acids, but also of essential fatty acids, which are necessary for the proper functioning of the body, and fat-soluble vitamins A, D, E, and K. It is true that, as the fat content of animal origin and non-liquid vegetable fats in the diet increases, both the contents of unfavorable-for-health saturated fatty acids, which are taken into account in assessing the Nutri-Score index, and the energy value of the diet increase. Since the Nutri-Score does not take into account the presence of fatty acids beneficial to the body in a product and only takes into account fatty acids that are potentially harmful, it may be that products with a higher content of essential fatty acids may receive a lower score just by having a higher total fat content and higher energy value, when, in fact, they would be a desirable component of the diet (Table 5). This difference will generally apply to fatty and lean fish, where the former will be rated lower despite providing more unsaturated fatty acids, including those of the omega-3 family, which are underrepresented in the Western diet, and vitamin D per serving.

### 4.6. Nutri-Score vs. Selected Nutrients

The Nutri-Score index does not take into account many health-promoting nutrients, as already discussed in the example of vitamins and minerals, and the presence of fruits and vegetables in the algorithm does not solve this problem. There are numerous bioactive ingredients in products today that significantly affect their nutritional value. 

While avoiding selected products, which, when consumed in reasonable amounts even daily, do not pose a health risk to the consumer, limiting their intake can lead to a reduction in the intake of certain nutrients that have health-promoting effects. Milk and dairy products, for example, are a source of conjugated linoleic acid dienes (CLA), which have anti-cancer and anti-atherosclerotic effects in addition to reducing fat synthesis and having a preventive effect against diabetes [47]. Thus, limiting products containing them will deplete the diet of an important nutrient. Additionally, products dedicated to athletes will often receive inferior gradation, and thus, may cause this group of consumers to limit their intake as well. An athlete without proper nutritional training is not significantly different from a typical consumer. The avoidance of high-carbohydrate products by this group, due to their likely low Nutri-Score, may consequently translate into decreased glycogen resynthesis or a lack of proper post-training recovery, and in turn, decreased physical performance and impaired athletic performance.

### 4.7. Nutri-Score vs. Food Market and Interpretation Errors

Within the European Union, the food labeling system in member countries is well regulated. However, legal standards allow for the introduction of additional markings in graphic form. At the same time, it is worth noting that additional indications of the nutritional value index of food are currently not mandatory, and manufacturers are free to use such an indication on the packaging or not. Consequently, such a system is introduced by a food manufacturer only when it is actually profitable for them to do so, i.e., when a product would receive a higher rating than a potential competitor’s product. A simple calculation is that a manufacturer will not be inclined to include a declaration of its inferior quality and nutritional value on the product’s packaging compared to a competing product. On the other hand, the absence of a Nutri-Score declaration on a product’s nutritional value next to a competing product with such a declaration may cause the consumer to reasonably doubt that a manufacturer who does not include such a declaration is not withholding information that its product is of inferior quality, when its nutritional value may be better, with only a Nutri-Score lower or the same compared to similar products.

The question of a consumer’s ability to spend money on Nutri-Score-labeled products has already been the subject of research. In their study, Mora-Garcia et al. (2019) showed that providing information on Nutri-Score resulted in 21% more spending on better-rated products. The results of the study indicated that participating customers were also 10% more likely to purchase a better-rated product than control customers, and Nutri-Score information increased store sales [48]. Nutri-Score labeling may thus become a marketing tool to increase revenue for selected food manufacturers, especially those whose products may be reformulated. Such treatments seem beneficial from the consumers’ point of view, as they get a product with a better composition. However, it should be noted that reformulation is possible for processed and ultra-processed foods. In the case of unprocessed products, there is little or no possibility of changing their composition. Ultimately, the consumer may get the message that processed and even ultra-processed foods are better than unprocessed or minimally processed foods. This may increase the consumption of highly processed foods, which are usually harmful to health.

### 4.8. Nutri-Score as a Tool to Justify the Choice of Selected Food Products

Consumer preferences always play one of the main roles in choosing a food product. It is well known that products characterized by sweet and/or salty flavors are more palatable and sensory-appealing to the consumer. The problem becomes when “unhealthy” products receive a high Nutri-Score gradation and the consumer uses this product rating system to justify potentially poor food choices, which, in the long run, may serve to perpetuate poor eating habits. As an example, products that, when consumed in excess, may prove to be detrimental to the consumer have received a better Nutri-Score rating simply by increasing one of the nutrients, such as protein or dietary fiber, while comparable products without the addition of these ingredients have received a worse rating. A prime example is the comparison of high-protein strawberry ice cream (Nutri-Score B) and vanilla cream cheese (Nutri-Score C). A potential consumer, looking at both packages and evaluating the products solely on the basis of the Nutri-Score, may come to believe that, in fact, the ice cream is healthier than the cheese. A higher product rating is predominantly due to the increased protein content of the product. If a consumer believes that ice cream is healthier than cream cheese, then in the future, remembering this choice, they may make the assumption that other ice creams without the Nutri-Score label would receive a similar rating, and thus would also be healthier than cheese. This, in turn, may lead to an over-consumption of ice cream that actually has a worse Nutri-Score if it does not contain a high-protein additive, deliberately introduced into the product formulation by the manufacturer in order to raise the rating in the Nutri-Score system. 

What should be further emphasized is that Nutri-Score refers to the nutritional value of a product per 100 g of that product. As presented earlier, it is highly likely that a consumer will choose and consume a product with a higher Nutri-Score gradient, but consuming 500 g of ice cream with a Nutri-Score B will provide significantly more simple sugars and saturated fatty acids than consuming 150 g of vanilla cream cheese. Failure to adjust the Nutri-Score to the portion habitually consumed and to the portion contained in the package once again gives the misconception that a product with a higher gradient is “healthier and safer” than a product with a lower gradient and can be consumed in more or any amount, without controlling for the effect on dietary energy value and saturated fatty acid, sugar, or salt content.

### 4.9. Regional and Organic Food in Evaluation by Nutri-Score System

Another aspect is the uniqueness of the composition of the food product. Regional and traditionally produced products have a unique composition, production method, and taste, which greatly enhances their culinary value. However, they may contain higher amounts of saturated fatty acids and salt, which translates into poorer gradation using the Nutri-Score system. The unique taste and culinary value, however, result in a much higher price compared to competing products, and all of these components combined translate into occasional consumption of such products, which are additionally consumed in small quantities, often less than 100 g. This also results in a low intake of nutrients negative to health, and despite the unfavorable gradation, these products do not pose a significant risk to consumer health, especially when consumed occasionally. The failure to take into account the usual amounts in which products are consumed is a drawback of this indicator, as has been repeatedly pointed out.

Organic products also have unique compositions and properties. Organically produced food is strictly regulated by law. The organic food production system takes a strict approach to the use of synthetic pesticides, the enrichment of feed, the use of excess drugs, and the possibility of using food additives in the product. Any modification of organic food production is impossible or very limited. Consequently, the composition and nutritional value of the products are very difficult to modify. Organic foods are less contaminated by pesticides and surface plant protection products [49,50,51,52,53], and heavy metals, and are richer in selected nutrients, such as antioxidants [54,55,56,57,58,59], n-3 polyunsaturated fatty acids [60,61,62], and CLA [62]. However, these values, although important from a health perspective for the consumer, are not taken into account in any way when evaluating using the Nutri-Score index. In addition, due to strict legal standards, it may be impossible or difficult to modify the composition of these foods (for the parameters used in the Nutri-Score assessment). This means that nutritionally valuable products may receive an inferior Nutri-Score simply because they do not have added protein (which increases Nutri-Score scores) or are not made as reduced-sugar or artificially sweetened products (which in turn translates into lower Nutri-Score scores). Thus, it may give the false perception that organic food is of lower quality than food produced using conventional methods despite the fact that there is no evidence to support such a claim.

## 5. Polemic against the Allegations

Some of the issues raised above that criticize the Nutri-Score system have already been discussed. Some issues have been raised, and the system’s co-authors have had the opportunity to respond to the issues mentioned.

In 2018, Julia et al. [63] addressed in a commentary the issue of converting the Nutri-Score to 100 g/mL of product, instead of the usual portion consumed. In their paper, the authors point out that the promoted competing Evolved Nutrition Label (ENL) system, which was developed by corporations on the basis of the MTL system, is just converted per serving of consumption, instead of per 100 g of product. Addressing these allegations, the authors stressed that the MTL system has been repeatedly compared to the Nutri-Score system in numerous studies conducted in France and has proven to be less functional in consumer evaluation. Despite the authors’ pertinent observation that the Nutri-Score system is a system that shows itself in a better light in studies than competing systems, this still does not explain the higher intake of selected negative nutrients with the conversion method used by the algorithm—per 100 g/mL of product. As demonstrated earlier, a consumer choosing a more favorably labeled product in a larger package will consume more energy, saturated fatty acids, and simple sugars as a derivative of the amount consumed than if they consumed a counterpart with a worse rating but in a smaller package. 

In both that paper and another from the same year [64], the authors emphasized the attempts to counter food manufacturers and entrepreneurs and the positive effects of the Nutri-Score. Regardless of the arguments raised, however, long-term studies with strong scientific evidence are needed to show that the Nutri-Score system will not produce negative health effects in certain population groups. While the idea and sophistication of Nutri-Score labeling are commendable and have positive results, the system does not take into account many factors that can affect public health. Therefore, if there is a risk of negative consequences, appropriate research is needed to minimize this risk.

Another paper addressed the allegation that the Nutri-Score system does not take into account the content of food additives and pesticides [65]. The authors emphasized that none of the currently existing FoP nutritional indices takes into account the presence of pesticides and food additives, and that the multiplicity and prevalence of additives in food products and ubiquitous exposure to pesticides make it impossible to adapt these factors to a functional labeling system. One has to agree with the authors on the main part of the opinion. First of all, referring to pesticides, due to the many possibilities of the contamination of food with pesticides and the potential likelihood of contamination only in specific batches of the product (and not in the product in general), it is not possible to create and apply such an algorithm in practice. This would additionally require changing the grading system for each separate batch of food product, which would additionally be labor- and cost-intensive. Another issue, however, is the need for manufacturers to eliminate food contamination, regardless of the presence of any indicator of the food’s nutritional value, and it is the responsibility and duty of the manufacturer to prevent such contamination. On the other hand, it should be pointed out that additives approved for food production are used for a specific purpose, to not only improve the attractiveness of the product, but often to fix the product, protect it from external factors (e.g., biological), or even enrich it with certain ingredients (e.g., vitamins). Thus, the negative impact of additives on consumer health cannot be approached unequivocally.

In addition, food additives are used in strictly defined amounts, which have been developed on the basis of numerous studies and labeled as safe for health. The problem is not the periodic consumption of a single product containing a significant amount of additives, but the frequent and irresponsible consumption of highly processed foods that contain a significant amount of frequently duplicated additives. The Nutri-Score system obviously cannot fully address the issue of additives in its evaluation of a food product, but it should certainly promote fresh/unprocessed and low-processed foods, and this is, however, lacking in this system.

## 6. Summary and Conclusions

Nutri-Score is a food nutritional index developed in France, which was designed to evaluate and rank food products to help the consumer make more informed and healthy food choices. Numerous studies have indicated that, indeed, the Nutri-Score system, as judged by the consumer, is easy to interpret and user-friendly.

Despite the enthusiasm for this labeling system, one is puzzled by the lack of thought given to its limitations in assessing the nutritional value of food and the disconnect between the information provided and the nutrient content not included in the algorithm, the size of the packages, the portion of the product traditionally consumed, the amount consumed within the dietary recommendations, the role of particular products in the diet, the degree of processing of the food, how the product is manufactured, the use of food additives, and some of the dietary recommendations.

Like any system, the Nutri-Score is not free of flaws. As the literature shows, the Nutri-Score may not promote a lower intake of simple sugars and a lower energy value of the dietary ration. In addition, misinterpretation of this index may lead to dietary restriction of foods rich in important nutrients, such as fatty marine fish, juices and nectars, cereal products with a low glycemic index, and products that affect the quality of the diet, such as traditionally produced regional and organic products. The Nutri-Score can also become a tool for promoting ultra-processed foods, in which the composition of individual ingredients included in the algorithm for evaluation is modified.

If the Nutri-Score system is to become a widely used front-of-pack label, it should be refined, take into account food ingredients that have a positive impact on consumer health, and pay attention to the issues raised above in order to perform its function properly. Future research should focus on assessing the risk of creating inappropriate behavior and dietary errors, including the elimination of selected product groups, by consumers, and the possibilities of preventive measures. The easy-to-interpret designation used in the Nutri-Score index by consumers is an advantage but also a disadvantage at the same time. It seems that by using a simple designation, the essential function it should perform is lost at the same time, i.e., to help match the diet with the recommendations of proper nutrition. Repeatedly emphasized by the system’s creators, its favorable reception by consumers actually testifies to good recognition of the universally accepted color code, where green means “safe” and red means “danger”. However, whether this will translate into better choices by consumers when buying food in the future remains an open question that needs to be investigated. Research should also focus on the possibility of improving the Nutri-Score algorithm to eliminate the aforementioned flaws and lead to the correct communication of nutritional information about a food product to the consumer.

## Figures and Tables

**Table 1 nutrients-14-04843-t001:** Comparison of nutrient contents of commercially available cream cheeses by package size.

	Cream Cheese XXL, Brand “X”		Cream Cheese, Brand “Y”	
Nutri-Score	B		C	
Weight (g)	100 (conversion)	200 (package)	100 (conversion)	150 (package)
Energy (kcal)	102	204	118	177
Protein (g)	5.3	10.6	4.8	7.2
Saturated fatty acids (g)	1.9	3.8	1.9	2.85
Sugars (g)	12.6	25.2	16	24
Fiber (g)	0	0	0	0
Salt (g)	0.08	0.16	0.1	0.15

**Table 2 nutrients-14-04843-t002:** Comparison of nutrient contents of commercially available cheeses by package size.

	Feta-Type Salad Cheese		Camembert Cheese Green Pepper	
Nutri-Score	D		D	
Weight (g)	100 (conversion)	55 (suggested serving)	100 (conversion)	30 (suggested serving)
Energy (kcal)	227	125	357	107
Protein (g)	10	5.5	17	5.1
Saturated fatty acids (g)	10	5.5	20	6.1
Sugars (g)	2.8	1.5	<0.05	<0.05
Fiber (g)	0	0	0	0
Salt (g)	3.5	1.9	1.7	0.51

**Table 3 nutrients-14-04843-t003:** Nutritional value of selected beverages (per 100 mL) and their Nutri-Score.

	Black Currant Nectar	Classic 100% Apple Juice	Cola-Type Drink Sweetened with Sweetener
Energy (kJ)	195	187	0.8
Protein (g)	<0.5	0	0
Saturated fatty acids	0	0	0
Sugars (g)	11	11	0
Fiber (g)	0	0	0
Salt (g)	0	0	0
Nutri-Score	E	C	B
**Vitamin C * (mg)**	**23.4**	**2.3**	**0**
**Iron * (mg)**	**1**	**0.2**	**0**
**Vitamin A * (mcg)**	**5**	**2**	**0**
**Vitamin E * (mg)**	**0.25**	**0.01**	**0**
**Bioactive compounds**	**Anthocyanins, polyphenols: anthocyanins, phenolic acid, flavonols, proanthocyanins**	**Polyphenols: catechins, rutin, phlorizin, chlorogenic acid**	**0**

* Based on “Tables of food composition and nutritional value” (Kunachowicz et al., 2018) [25]. Ingredients in bold are not included in the Nutri-Score.

**Table 4 nutrients-14-04843-t004:** Selected nutritional value of selected rice (per 100 g) and their Nutri-Score.

	White Rice	Parboiled Rice	Basmati Rice	Jasmine Rice
Energy (kJ)	1441	1458	1458	1469
Protein (g)	8.2	8.3	6.5	7.6
Saturated fatty acids	0.15	0.15	0.2	0.2
Sugars (g)	0.6	0.4	0.2	0.2
Fiber (g)	1.6	2.2	2.4	1.5
Salt (g)	0	0	0	0
Nutri-Score	A	A	A	A
Glycemic index	80	50	50	68

**Table 5 nutrients-14-04843-t005:** Selected nutritional value of selected fish (in 100 g) and their Nutri-Score index.

	Cold-Smoked Salmon	Fresh Salmon *	Fresh Salmon Trout *
Energy (kJ)	704	886	678
Protein (g)	19	18.4	18.1
Saturated fatty acids	1.62	2.6	1.5
Sugars (g)	0	0	0
Fiber (g)	0.7	0	0
Salt (g)	3.0	0.1	0.12
Nutri-Score	D	B	A

* Products did not have a Nutri-Score index assigned by the manufacturer. Nutri-Score values were calculated using the instructions provided by the system’s authors.

## Data Availability

Not applicable.

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
