# Peer review of "Fantastic Foods and Where to Find Them—Advantages and Disadvantages of Nutri-Score in the Search for Healthier Food"

_nutrients, 2022, doi:10.3390/nu14224843_

Round 1

Reviewer 1 Report

The review is well-detailed and clearly highlights the points of disagreement with the current Nutri-Score system, underlining how it can induce the consumer to make choices that often do not correspond to the healthiest.

I only have comments regarding the fluidity of the reading.

Title: it is certainly funny and modern, but, in my opinion, it does not reflect the intent of the review: here is reported a critical analysis of the Nutri-Score system, and this is not predictable from the title.

Lines 53-66:  the description of the system is convoluted and not understandable at first reading. Being the subject of criticism, it should be reformulated to make the Nutri- Score evaluation system clearer.

Lines 96-97: It should be specified how long this Nutri-Score system has been in use (also in the introduction).

Lines 128-129: Since the 5-CNL system is previously explained, the criterion of English Multiple Traffic Lights (MTL) should also be explained.

Lines 234 to 313: I find the discussion on the benefits of consuming fruit juices too loquacious and detailed; this part should be shortened; the concept is clear even without listing all the benefits in detail.

Table 5: The intent of the table is not fully clear. If only the first product has a Nutri-Score index assigned from the manufacturer, the other two by whom were they assigned and according to what criteria? It should be clarified.

Lines 386-390: This seems to me an important information that cannot appear only at this point. It should be anticipated in the global introduction.

Reviewer 2 Report

This paper critically analyzed the Nutri-Score system as a Front-Of-Pack (FoP) labeling system.  I assume that this study is very meaningful and instructive, blow are some concerns need to be explained or improved.

1. Introduction relevant to nutri-score system is suggested to be further improved, including How wide the application range, how much influence, how unscientific, which would be further improve the meaning of this study.

2. The "various health effects of multiple fruit juices" part, could you supplement the specific active ingredients in different fruit juices, which would be more convincing and scientific.

3. "4.3. Nutri-Score vs. fruit and vegetable juices" section,  the expressions are suggested to be further rearranged and revised.

4. Since those disadvantages of Nutri-Score were raised, could the authors propose any effective improvement measures or implementation plans, and feasibility analysis?

5. The title "Fantastic foods and where to find them – is Nutri-Score helpful?" was suggested to be modified, since the authors demonstrate that the Nutri-Score showed both advantages and disadvantages.

6. There is a serious lack of references in the past three years. It is suggested to supplement more than 20 related research results in recent years.
